



# Distribution characteristics of summer precipitation raindrop spectrum in Qinghai−Tibet Plateau

Fuzeng Wang[1,2], Yao Huo[1], Yaxi Cao[1], Qiusong Wang[1], Tong Zhang[2], Junqing Liu[3], Guangmin Cao[4]

[1]College of Electronic Engineering, Chengdu University of Information Technology, Chengdu 610225, China

[2]Key Laboratory of Land Surface Process and Climate Change in Cold and Arid Regions, Chinese Academy of Sciences, Lanzhou 730000, China

[3]Weather Modification Center for Tibet, Lhasa 850000, China

[4]Heilongjiang Meteorological Data center, Harbin 150000, China

*Correspondence to*: Guangmin Cao(ccgm909@163.com)

**Abstract:** To enhance the precision of precipitation forecasting in the Qinghai−Tibet Plateau region, a comprehensive study of both macro− and micro−characteristics of local precipitation is imperative. In this study, we investigated the particle size distribution, droplet velocity, droplet number density, $Z$ (Radar reflectivity) − $I$ (Rainfall intensity) relationship, and Gamma distribution of precipitation droplet spectra with a single precipitation duration of at least 20 minutes and precipitation of 5 mm or more at four stations (Nyalam, Lhasa, Shigatse, and Naqu) in Tibet during the recent years from June to August. The results are as follows: (1) In the fitting relationship curve between precipitation raindrop spectral particle size and falling speed at the four stations in Tibet, when the particle size was less than 1.5 mm, the four lines essentially coincided. When the particle size exceeded 1.5 mm, the speed in Shigatse was the highest, followed by Lhasa, and the speed in Naqu was the lowest. The falling speed of particles correlated with altitude. (2) The diameter of the six microphysical features at the four stations increased with altitude. (3) The $Z−I$ relationships at the four stations exhibited variations. Owing to the proximity in altitude between Lhasa and Shigatse, as well as between Nyalam and Nagqu, the coefficients $a$ and index $b$ in the $Z−I$ relationships of the two groups of sites were relatively similar. (4) The fitting curves of the M−P and Gamma distributions of the precipitation particle size at the aforementioned four stations are largely comparable. The M−P distribution fitting exhibits a slightly better effect. The parameter $\mu$ in Gamma distribution decreases with the increase of altitude, while $N_0$ and $\lambda$ in M−P distribution show a clear upward trend with altitude.



**1. Introduction**


The microphysical processes of cloud and precipitation over the Qinghai−Tibet Plateau significantly
differ from those in low−altitude regions due to the high average altitude and complex, changeable terrain,
resulting in a strong ground heating effect. Due to the terrain's influence, the plateau area has a limited
number of observation stations, leading to a scarcity of precipitation records. Based on three atmospheric
scientific experiments conducted over the Qinghai−Tibet Plateau, convective clouds exhibit high activity,
although the precipitation intensity is moderate(Li et al., 2014; Jiang et al., 2002; Xu et al., 2006; Li el
al., 2001). In the central part of the Plateau, severe convective clouds constitute 4% to 21%, with
cumulonimbus clouds representing 21%. Additionally, the frequency of severe weather, such as
thunderstorms and hail, surpasses that in other regions. In the majority of Qinghai−Tibet Plateau areas,
convective cloud precipitation constitutes over 90% of the total (Chang and Guo, 2016). Particularly
during the rainy season, convective processes are frequent with smaller horizontal scales, weaker
intensities, and shorter durations. Due to observational constraints, short−term tests and satellite data
(e.g., TRMM, CloudSat, and Aqua) are employed to investigate Tibetan Plateau precipitation, with a
focus on liquid precipitation characteristics, including seasonal and diurnal variations and convective
activity's liquid drop spectrum inversion(Ruan et al., 2015; Liu et al., 2015; Xiong et al., 2019; Zhang et
al., 2018). The scarcity of observational data on cloud precipitation's physical processes in the
Qinghai−Tibet Plateau results in limited studies on microscopic parameters' characteristics. The recent
installation of a laser raindrop spectrometer enables a comprehensive understanding of the plateau's
precipitation microphysical parameters through the study of raindrop spectral parameters and distribution
characteristics in various regions.
Some studies have explored the spectral characteristics of raindrops over the Tibetan Plateau. Yu Jianyu
et al. and Shu Lei et al. conducted analyses on the raindrop spectrum characteristics of various clouds in
the Naqu and Yushu regions of the Qinghai−Tibet Plateau(Yu et al., 2020; Shu et al., 2021). Li Shanshan
et al. investigated raindrop spectral characteristics at different elevations on the eastern slope of the
Qinghai−Tibet Plateau. They discovered that the average spectrum of raindrop number concentration at
various elevations conforms to the Gamma function distribution. Moreover, light precipitation and heavy
precipitation exhibit distinct raindrop spectral characteristics(Li et al., 2020). The aforementioned
research was conducted in Naqu and Yushu areas in the Qinghai−Tibet Plateau, as well as the west



Sichuan Plateau area. However, there is a limited number of studies on the spectral characteristics and
distribution rules of cloud precipitation raindrops in various regions of the Tibetan Plateau. The analysis
of raindrop spectrum characteristics in the Naqu region, as mentioned earlier, was conducted only during
the summer months from June to August 2014. In this study, we used raindrop spectrum data from the
Naqu region spanning 2017 to 2020, building upon and extending previous research. We analyzed the
temporal variation of the raindrop spectrum in convective cloud precipitation across various regions and
examine differences in raindrop spectra among these regions. We conducted a systematic analysis of
raindrop spectrum data associated with moderate rain from four stations with varying altitudes,
longitudes, and latitudes. We compared and analyzed the differences in drop spectrum characteristics
among these four stations, which is of great significance for enhancing the scientific understanding of
precipitation's influence in the plateau region.
The objective of this study is to enhance the understanding of raindrop spectrum characteristics at various
elevations of the Tibetan Plateau. The findings of this study will establish a foundation for
comprehending precipitation characteristics and improving precipitation forecasts at diverse elevations
of the Tibetan Plateau. This study is structured as follows: Data sources and research methods are
described in Section 2. The analysis results are presented in Section 3 while the conclusion and discussion
are provided in Section 4.
**2. Data and methods**
**2.1. Data collection**
The data obtained for this study consist of raindrop spectrum data from four meteorological stations (i.e.,
Nyalam, Lhasa, Shigatse, and Naqu) in Tibet. Owing to its unique climate environment, snowfall occurs
time to time from September to May. Data from June to August is selected to analyze the precipitation
raindrop spectrum process in this study. The precipitation data selection criteria include a precipitation
process duration exceeding 20 minutes and a single precipitation process with rainfall greater than 5mm.
As the frequency of convective clouds in most areas of the Qinghai−Tibet Plateau exceeds 90%, all
collected samples are categorized as convective clouds in this paper. Table 1 displays the longitude,
latitude, altitude, and sample numbers of the four stations. Figure 1 illustrates the geographical
distribution of the four sites. The four stations cover a broad area of central Tibet from south to north,



making the results representative.

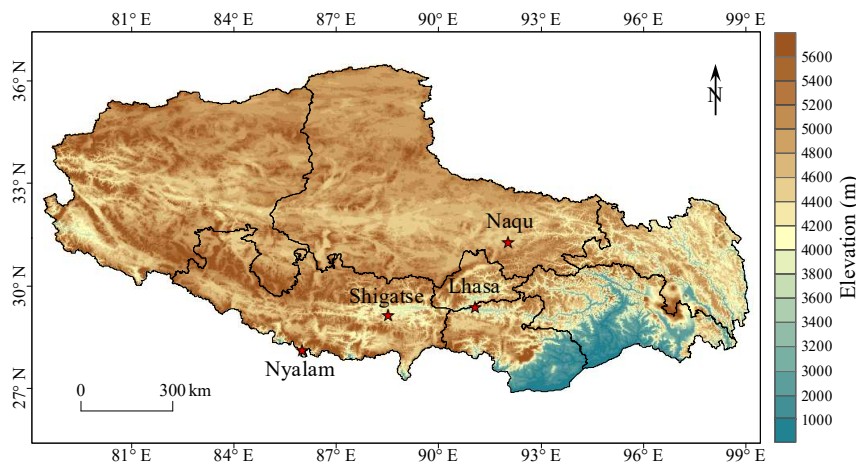


**Figure 1: Station distribution and the surrounding terrain**
**Table 1: Coordinates, elevation, sampling periods, and sample sizes of the four sites.**

| Station | Longitude | Latitude | Elevation | Sampling period | Sample size |
|---------|-----------|----------|-----------|-----------------|-------------|
| Nyalam | 85.58° E | 28.11° N | 4519 m | 2017−2019 | 11579 |
| Lhasa | 91.08° E | 29.40° N | 3653 m | 2017−2018 | 8364 |
| Shigatse | 88.53° E | 29.15° N | 3910 m | 2017−2018 | 14237 |
| Naqu | 92.04° E | 31.29° N | 4560 m | 2017−2020 | 5630 |

**2.2. Quality Control and Quality Assurance (QA/QC)**
The Parsivel2 raindrop spectrometer features 32 particle size measurement channels and 32 particle
velocity measurement channels. The particle size measurement range is 0.062−24.5 mm, and the particle
velocity measurement range is 0.05−20.8 m s$^{-1}$, with a sampling time of 60 s. In comparison to the
previous Parsivel raindrop spectrometer model, the Parsivel2 raindrop spectrometer utilizes infrared light
as its light source. This change reduces the interference of visible light, resulting in significant
advancements in the measurement of raindrop size and rainfall. Following the sampling principle of the
raindrop spectrometer, the instrument records the particle size and particle speed of all particles passing
through the sampling surface. To mitigate the influence of sand and dust particles, it is imperative to
control the quality of the fundamental data.
Atlas(Atlas et al., 1973) discovered a relationship between the final falling velocity of particles and the
particle diameter. In an ideal windless environment, the formula for the final falling velocity of particles



is:

$$\begin{cases} v=0, & x<0.03 \\ v = 4.323 \times (x-0.03), & 0.03 \le x \le 0.6 \\ v = 9.65 - 10.3 \times e^{-0.6x}, & x > 0.6 \end{cases}$$

104                                                                                          (1)

where $x$ represents the particle diameter in mm, and $v$ represents the final falling velocity of the particle
in m s$^{-1}$.
Kruger and Krajewski(Kruger and Krajewski, 2002) proposed a method to mitigate the dispersion of
velocity over large samples, building on the study by Atlas. Initially, the final falling velocity was
calculated based on the particle diameter and final velocity formula, and subsequently, a threshold value
was set for elimination. The formula is expressed in Equation 2.
$$\left| v_{measured} - v_A \right| < 0.4 v_A \qquad (2)$$
where $v_{measured}$ represents the final velocity measured by the raindrop spectrometer, and $v_A$ is the final
velocity calculated using the final velocity formula. If the relative error falls within the specified
threshold range, the data will be retained.
Previous studies have highlighted that the distribution of raindrop spectrum exhibits distinct
characteristics influenced by geographical environment and topography. Hence, utilizing the same
calculation formula across different areas for raindrop spectrum elimination is likely to introduce
significant errors. Therefore, we utilized historical data from a raindrop spectrum site to localize the
parameters identified in the study by Atlas and incorporates them into the formula for particle elimination.
Simultaneously, due to deformation occurring in raindrops during descent, the raindrop spectrum data
undergoes deformation and correction after quality control. Battaglia() defined the axial ratio ($ar$) as the
ratio of radial and transverse lengths of raindrop particles. Particles with a particle size less than 1 mm
are defined as spherical. The axial ratio is defined as $ar = 1.075 - 0.075Deq$ for particles with a particle
size of 1−5 mm, where $Deq$ is the equivalent precipitation particle diameter, and $ar = 0.7$ for particles
with a particle size greater than 5 mm.
**2.3. Raindrop spectrum parameters**
The number density of the precipitation raindrop spectrum is defined as the total number of particles per
unit volume(Shi et al., 2008).



$$N(D) = \sum_{i=1}^{32} \sum_{j=1}^{32} \frac{n_{ij}}{A \cdot \Delta T \cdot V_j}$$


(3)

where N(D) is the number density parameter, in units of mm−1 m−3; nij represents the number of
raindrops with the diameter of the i−th particle and the velocity of the j−th particle; A is the sampling
base area of the raindrop spectrometer (5400 mm2); ΔT is the sampling time (60 s); Vj is the velocity
value of the sampled particle, in units of m s−1.
The average diameter is calculated as the sum of the diameters of all raindrops per unit volume divided
by the total number of raindrops, and the formula is given by equation 4.

$$D_l = \frac{\sum_{i=1}^{32} N(D_i)D_i}{\sum_{i=1}^{32} N(D_i)}$$


(4)

The weighted average diameter represents the average diameter of the weighted mass of all particles per
unit volume relative to the total mass of particles, measured in mm. The formula is expressed in equation

139  5.

$$D_m = \frac{\sum_{i=1}^{32} N(D_i)D_i^4}{\sum_{i=1}^{32} N(D_i)D_i^3}$$


(5)

where Di represents the diameter of the i−th particle, and N(Di) represents the particle number density
of the i−th particle diameter.
Precipitation intensity refers to precipitation per unit time (per hour), measured in mm h−1. The formula
is given by equation 6.

$$I = \frac{6\pi}{10^4} \sum_{i=1}^{32} D_i^3 V(D_i) N(D_i)$$


(6)

The radar reflectivity factor is the sum of the backscattering area of all particles per unit volume,
measured in mm−6 m−3. The formula is expressed in equation 7.

$$Z = \sum_{i=1}^{32} N(D_i)D_i^6$$


(7)

The observed raindrop spectrum is discrete, and the double−parameter index, namely M−P distribution,
can be used to simulate the raindrop particle size distribution. The formula is given by equation 8.



$$N(D) = N_0 \times \exp(-\lambda D) \qquad (8)$$

where N0 is a number density parameter, measured in mm−1 m−3. $\lambda$ is a size parameter, measured in
mm−1.
However, this distribution pattern has some errors compared with actual observation data when
describing small and large raindrops. Therefore, Ulbrich and Atlas proposed a modified raindrop particle
size distribution pattern. They treated the raindrop spectrum distribution as a Gamma distribution to
correct the distribution pattern between small and large raindrops.
In this case, the raindrop particle size distribution follows the Gamma distribution with three
parameters(Carlton and David, 1984). The formula is given by equation 9.
$$N(D) = N_0 \times D^{\mu} \times \exp(-\lambda D) \qquad (9)$$

where $\mu$ is a dimensionless parameter referred to as the shape factor. When $\mu$ is greater than 0, the curve
exhibits an upward curvature; when $\mu$ is less than 0, the curve displays a downward curvature. When
$\mu=0$, it corresponds to an M−P distribution.
Zhang(Zhang et al., 2003) pointed out a binomial relationship between $\mu$ and $\lambda$ when studying the $\mu-\lambda$
relationship of precipitation in Florida:
$$\lambda = a\mu^2 + b\mu + c \qquad (10)$$

Ulbrich(Ulbrich, 1983) pointed out in his study that the $\mu-\lambda$ relation under Gamma distribution can be
expressed as:
$$D_m = \frac{4+\mu}{\lambda} \qquad (11)$$

Equation (11) shows that there is a relationship between the ratio of $\mu$ and $\lambda$ and the weighted average
diameter of mass. The Gamma distribution fit is typically applied to the observed raindrops distribution
N(D) using the least squares or order moments method. In this study, the least square method is employed
to fit the M−P and Gamma distributions.

**3. Result and discussion**

The average altitude of the Qinghai−Tibet Plateau is over 4000 m, and the terrain is complex and
changeable, resulting in varying microphysical characteristics of the raindrop spectrum. Therefore,
considering the unique conditions of the Qinghai−Tibet Plateau, the rain intensity calculated based on



the raindrop spectrum was categorized into five grades for calculation and analysis, as presented in Table
2. The results indicated that the mean value and standard deviation of the rain intensity at the same station
were generally proportional to the rain intensity, with slight fluctuations observed among individual
stations. The samples from the four stations in the range of 0.5−5 mm·h−1 were the largest, and the
obtained standard deviation values were all very small. This indicates a high consistency in rain intensity
distribution under weak rain intensity. In the interval of precipitation intensity greater than 20 mm h−1,
only two stations have samples, and one of the stations exhibits a large standard deviation. This reflects
a significant inversion error in raindrop spectrum for Nyalam during short−duration heavy precipitation.
**Table 2: Descriptive statistics of rainfall intensity at the four stations.**

|  | Range (mm·h−1) | Sample Size | Mean (mm·h−1) | Standard Deviation | Precipitation (mm) |
|---|---|---|---|---|---|
|  | 0.5−5 | 4047 | 2.16 | 1.21 | 146 |
|  | 5−10 | 1358 | 7.38 | 1.28 | 166.6 |
| Nyalam | 10−15 | 900 | 12.14 | 1.32 | 182.1 |
|  | 15−20 | 656 | 17.69 | 1.37 | 193.4 |
|  | >20 | 960 | 30.63 | 7.99 | 490 |
|  | 0.5−5 | 3245 | 1.8 | 0.94 | 97.4 |
|  | 5−10 | 180 | 5.87 | 0.77 | 17.6 |
| Lhasa | 10−15 | 50 | 12.1 | 0 | 12.1 |
|  | 15−20 | 0 | 0 | 0 | 0 |
|  | >20 | 0 | 0 | 0 | 0 |
|  | 0.5−5 | 7094 | 1.78 | 1.06 | 210.7 |
|  | 5−10 | 584 | 6.37 | 1.11 | 62.02 |
| Shigatse | 10−15 | 60 | 10.01 | 0 | 10.01 |
|  | 15−20 | 0 | 0 | 0 | 0 |
|  | >20 | 0 | 0 | 0 | 0 |
|  | 0.5−5 | 2389 | 3.27 | 1.5 | 130.1 |
| Naqu | 5−10 | 675 | 7.76 | 1.1 | 87.3 |
|  | 10−15 | 479 | 13.73 | 1.21 | 109.6 |





| | | | | |
|---|---|---|---|---|
| 15−20 | 372 | 19.65 | 1.4 | 121.8 |
| >20 | 120 | 21.6 | 1.5 | 43.2 |

**3.1. Precipitation particle size, speed and rainfall intensity contribution rate distribution**

Figure 2 represent the mean precipitation values across the four stations. The canvas is divided into several rectangular areas defined by the coordinates of the horizontal and left axes, and the color code is applied to them. Each rectangular area represents a specific particle diameter and velocity. Figure 2 reveals that the fitting curves of particle diameter distribution and final falling velocity at the four stations are approximately identical, and the final falling velocity increases with the particle diameter. Regarding particle number density, it is concentrated in the area with particle size less than 1 mm, and it decreases with the increase of diameter. Concerning the contribution rate of precipitation intensity, the four stations exhibit a multi−peak distribution, with peak diameters at 0.812 mm and 1.375 mm. In comparison with the precipitation process of convective clouds at low−altitude stations, the particle size spectrum width at the four stations on the Tibetan Plateau in this analysis was notably reduced, and the particle number density at the four stations with particle sizes greater than 3 mm was very low.

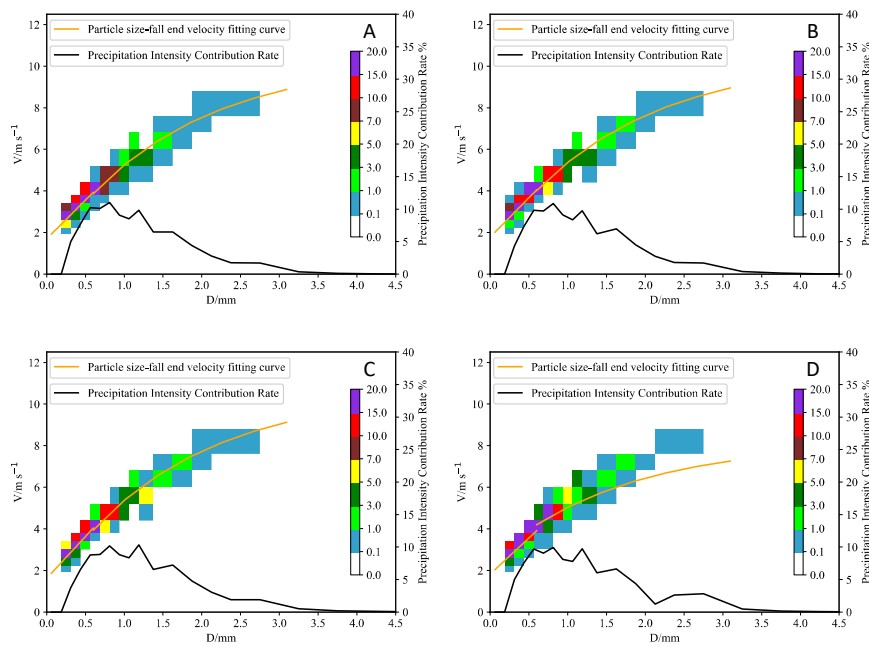





**Figure 2: The average spectrum of precipitation particle size, velocity, and contribution rate distribution of**
**precipitation intensity. The color bar represents the number density in units per m3. (A. Nyalam, B. Lhasa,**
**C. Shigatse, and D. Naqu).**
Figure 3 displays the fitting relationship between the particle size of the raindrop spectrum and the falling
speed at the four stations in Tibet. For particle sizes less than 1.5 mm, the particle size at the four stations
essentially aligns with the final falling speed. For particle sizes greater than 1.5 mm, the speed is largest
for Shigatse, followed by Lhasa, and Naqu has the smallest speed. However, under the same size, the
final velocities of particles at the four stations are greater than those in Guizhou, exceeding 2 m/s. This
may be attributed to the higher altitude of the four stations, which are over 3000 m above sea level. This
indicates that the high altitude of Tibet, due to thin air and low air pressure, results in decreased fall speed
of larger particles of the same size. However, particles at lower altitudes (Shigatse and Lhasa) exhibited
slightly higher speeds than those at higher altitudes (Nyalam and Naqu). This difference may be
attributed to the instruments at higher altitudes being closer to the clouds, leading to the detection of
raindrops before they interacted with each other. The fitting formulas for the v−D relationships at the
four sites (Nyalam, Lhasa, Shigatse, and Naqu) are given by Equations 12, 13, 14, and 15, respectively.
$$\begin{cases} v=0, & x<0.03 \\ v = 3.720 \times (x+0.456), 0.03 \leq x \leq 0.6 \\ v = 10.325 - 9.252 \times e^{-0.6x}, x > 0.6 \end{cases} \quad (12)$$

$$\begin{cases} v=0, & x<0.03 \\ v = 3.796 \times (x+0.468), 0.03 \leq x \leq 0.6 \\ v = 10.375 - 9.118 \times e^{-0.6x}, x > 0.6 \end{cases} \quad (13)$$

$$\begin{cases} v=0, & x<0.03 \\ v = 4.035 \times (x+0.401), 0.03 \leq x \leq 0.6 \\ v = 10.614 - 9.568 \times e^{-0.6x}, x > 0.6 \end{cases} \quad (14)$$

$$\begin{cases} v=0, & x<0.03 \\ v = 3.474 \times (x+0.524), 0.03 \leq x \leq 0.6 \\ v = 10.162 - 9.018 \times e^{-0.6x}, x > 0.6 \end{cases} \quad (15)$$





**Figure 3: The relationship between particle size and speed at four stations.**

The proportion of particle number density in raindrop spectrum and the contribution rate of precipitation are shown in Table 3 and Table 4, respectively.

**Table 3: Percentage of particle number density.**

|  | Particle diameter (mm) | | |
| --- | --- | --- | --- |
|  | 0−1 mm | 1−2 mm | 2−3 mm |
| Nyalam | 93.60 | 6.15 | 0.25 |
| Lhasa | 92.41 | 7.24 | 0.35 |
| Shigatse | 91.45 | 8.06 | 0.49 |
| Naqu | 91.89 | 7.52 | 0.59 |

**Table 4: Percentage of precipitation contribution rate.**

|  | Particle diameter (mm) | | |
| --- | --- | --- | --- |
|  | 0−1 mm | 1−2 mm | 2−3 mm |
| Nyalam | 55.63 | 37.32 | 7.05 |
| Lhasa | 54.60 | 38.16 | 7.24 |
| Shigatse | 51.12 | 40.49 | 8.39 |
| Naqu | 54.06 | 37.81 | 8.13 |

It can be observed from Table 3 that the number of precipitation particles with a distribution of 0−1 mm



constitutes the largest proportion, exceeding 91%, while the proportion of particles with a distribution of
more than 3 mm is comparatively smaller, being less than 0.6%. The proportion of precipitation intensity
below 1 mm constitutes over 51%, with other particles comprising less than 49%. The results indicate
that the contribution of precipitation intensity on the Tibetan Plateau is primarily concentrated in small
particles with a diameter less than 1 mm.
Simultaneously, it is observed that small particles below 1mm in Shigatse are smaller than those at other
stations, and particles above 3 mm are larger than those at the other three stations. In contrast to the
convective cloud precipitation in Zheng'an, Guizhou analyzed by Wang(Wang et al., 2020), where
convective cloud particles less than 1 mm account for 64.4%, the contribution rate to precipitation is only
17%; Additionally, it significantly differs from the rainstorm in Hainan analyzed by Mao(Mao et al.,
2020). Despite the proportion of less than 1mm being 82.7%, the contribution rate is only 18.2%, and
the rainstorm particle size spectrum in Hainan is remarkably wide. It is evident that the precipitation
characteristics of convective clouds on the Qinghai−Tibet Plateau exhibit a particularity, wherein the
diameter of precipitation particles is generally small, and the precipitation of small−diameter particles
constitutes a substantial proportion of the total precipitation.
**3.2. Microphysical characteristic parameters of precipitation**
Calculation of characteristic parameters such as diameter (Dm), average volume diameter (Dv), mode
diameter (Dd), dominant diameter (Dp), and medium diameter (Dnd) was conducted. Based on the
comprehensive analysis of the characteristic parameters in Table 5, the Dm size at Lhasa station with the
highest altitude (Naqu) is the largest, while the Dm size at the station with the lowest altitude (Lhasa) is
the smallest. The values at the stations in Lhasa and Shigatse, with intermediate elevations, fall between
the two extremes. The particle size at the Nyalam station, with a higher elevation, is also greater than
that at the station in Shigatse, which is at a lower elevation. Simultaneously, the diameters of other
features also increased with elevation, similar to Dm. Additionally, the differences in characteristic
diameters among the stations in Nyalam (4519m), Naqu (4560m), Lhasa (3653m), and Shigatse (3910m)
with similar altitudes are relatively small. The preceding analysis demonstrates a strong positive
correlation between altitude and these six microphysical characteristic parameters.
**Table 5: Microphysical parameters at the four stations.**

| Station | Dm | Dv | Dd | Dp | Dnd |
|---------|----|----|----|----|-----|



| Nyalam | 1.296 | 1.853 | 0.527 | 2.143 | 1.81 |
| Lhasa | 0.85 | 1.432 | 0.436 | 0.953 | 0.848 |
| Shigatse | 0.878 | 1.458 | 0.44 | 0.989 | 0.881 |
| Naqu | 1.302 | 1.961 | 0.565 | 2.18 | 1.883 |

**3.3. Z−I relation distribution**

Utilizing Formulae (6) and (7), the radar reflectivity (Z) and precipitation intensity (I) are calculated

independently, and the data undergo fitting. The results are depicted in Figure 4.

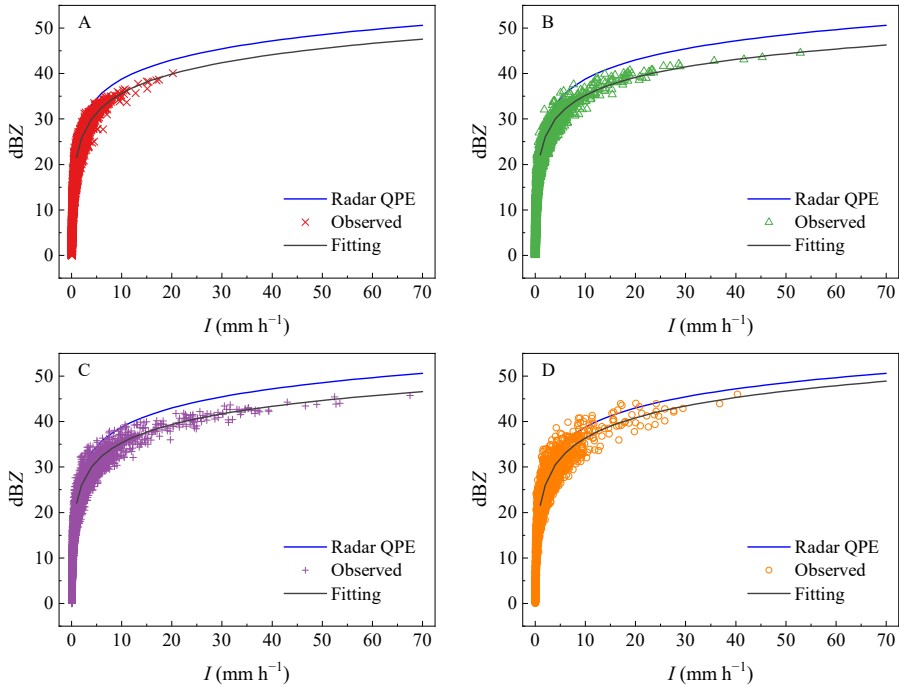

**Figure 4: The Z−I relationships at four stations. (A. Nyalam, B. Lhasa, C. Shigatse, and D. Naqu)**

Figure 4 reveals that the suggested reference relation $Z=300 \times I^{1.4}$ inaccurately predicts precipitation,

leading to an underestimation of precipitation intensity under identical radar reflectivity. With identical

radar reflectance, the precipitation intensity is highest in Lhasa, followed by Shigatse, while the smallest

precipitation intensity was observed in Naqu.

Table 6 shows the results of fitted Z−I relationships. Analyzing the altitude based differences in the Z−I

relationship, the a and b coefficients are similar for the station at 3653 m (Lhasa) and the station at 3910

m (Shigatse), while a and b for the station at 4519 m (Nyalam) and the station at 4560 m (Naqu) are



close. This observation indicates that the fitting parameter a is notably smaller, and the fitting parameter
b is larger for stations at higher altitudes.
**Table 6: Z−I relationship fitting results.**

|  | $Z = aI^b$ | | |
|---|---|---|---|
|  | Fitting | a | b |
| Nyalam | $Z=143.01×I^{1.41}$ | 143.01 | 1.41 |
| Lhasa | $Z=162.56×I^{1.31}$ | 162.56 | 1.31 |
| Shigatse | $Z=160.21×I^{1.33}$ | 160.21 | 1.33 |
| Naqu | $Z=143.81×I^{1.48}$ | 143.81 | 1.48 |

**3.4. Precipitation particle distribution fitting**
According to Formulas (8) and (9), the least squares method is applied to fit the M−P and Gamma
distributions of the mean raindrop spectrum of precipitation at the four stations. The results are presented
in Figure 5 and Table 7.

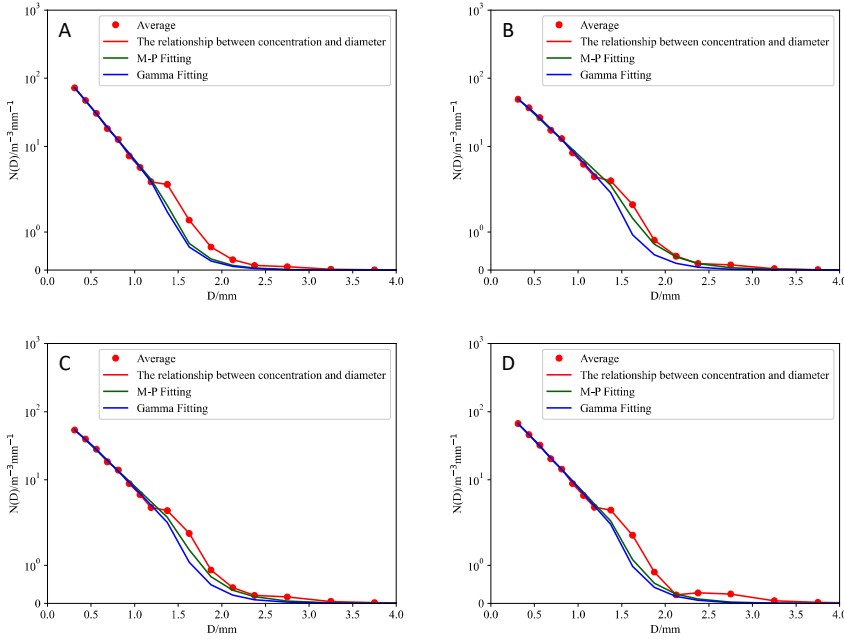


**Figure 5: M−P and Gamma istributions for precipitation (A. Nyalam, B. Lhasa, C. Shigatse, and D. Naqu).**
As indicated in Table 7, μ decreases with increasing altitude in the Gamma distribution. A smaller μ
corresponds to a wider raindrop spectrum, signifying a larger change in raindrop diameter with increasing



altitude. The raindrop diameter at higher altitudes is larger, corresponding to the precipitation
microphysical characteristics calculated in Table 5. Conversely, the fitting results of the M−P distribution
show that N0 and λ exhibit a clear increasing trend with height. In Figure 5, the abscissa represents
particle diameter, and the ordinate represents particle number density. The curve trends at the four stations
are relatively consistent. For Nyalam station, the M−P distribution is given by N(D)=218.78×e−3.53D,
and the Gamma distribution is N(D)=282.14×D0.15×e−3.82D. For Lhasa station, the M−P distribution
is N(D)=118.70×e−2.75D, and the Gamma distribution is N(D)=250.40×D0.43×e−3.56D. For Shigatse
station, the M−P distribution is N(D)=130.35×e−2.79D, and the Gamma distribution is
N(D)=216.08×D0.29×e−3.35D.    Finally,    for    Naqu    station,    the    M−P    distribution    is
N(D)=177.22×e−3.10D, and the Gamma distribution is N(D)=238.95×D0.17×e−3.44D. In the Gamma
distribution, two parameters, μ and λ, represent the curve shape factor and particle scale parameters,
respectively, as shown in Equation (9). According to Equation (10), the two parameters μ and λ for the
four stations are fitted with an analytical binomial relationship, and the coefficients are presented in Table

290    8.

**Table 7: Gamma fitting and M−P fitting results.**

|  | Gamma | | | M−P | |
|---|---|---|---|---|---|
|  | $N_0$ | $\mu$ | $\lambda$ | $N_0$ | $\lambda$ |
| Nyalam | 284.90 | 0.15 | 3.83 | 218.93 | 3.53 |
| Lhasa | 253.26 | 0.44 | 3.59 | 118.81 | 2.75 |
| Shigatse | 217.69 | 0.30 | 3.35 | 130.45 | 2.79 |
| Naqu | 240.91 | 0.18 | 3.45 | 177.34 | 3.10 |

**Table 8: μ and λ binomial parameters**

|  | $\lambda = a\mu^2 + b\mu + c$ | | |
|---|---|---|---|
|  | $a$ | $b$ | $c$ |
| Nyalam | 0.2816 | 1.2798 | 1.5074 |
| Lhasa | 0.1717 | 1.0589 | 1.3983 |
| Shigatse | 0.0221 | 1.1215 | 1.6002 |
| Naqu | 0.0155 | 1.2141 | 1.7599 |

It can be observed from Figure 6 that, although the four curves bend towards the lambda axis, the degree
of bending varies. The curves for Shigatse exhibit nearly straight curves, whereas the curves for Nyalam
and Naqu are more pronounced in their curvature towards the lambda axis. The μ−λ relationship varies
among the four stations, and this variation is associated with the mass−weighted diameter. Eq. (11)
indicates that when λ remains constant, a higher μ value corresponds to a greater mass−weighted average



diameter.

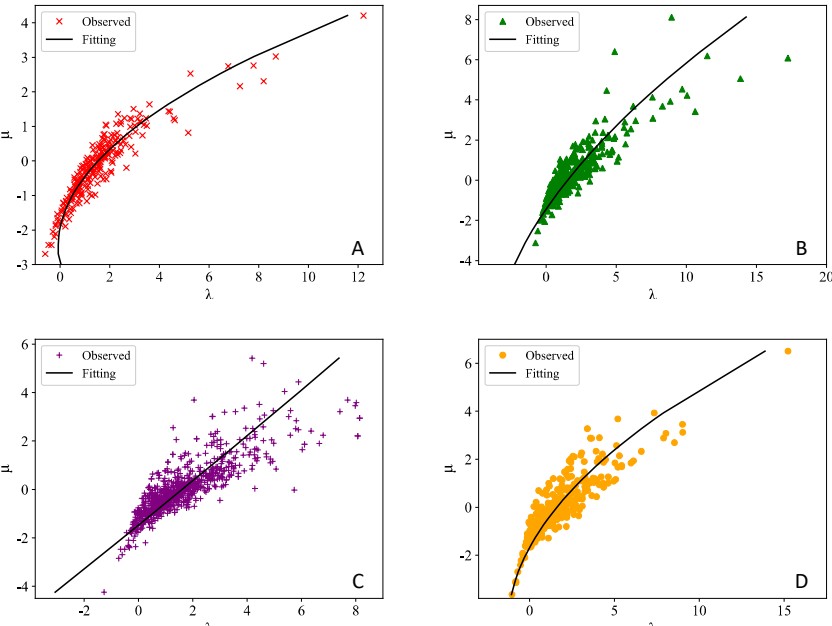


**Figure 6: μ−λ relationship (A. Nyalam, B. Lhasa, C. Shigatse, D. Naqu).**
**4. Conclusions**
In this study, we conducted a statistical analysis of raindrop spectrum data above moderate rain at four
sites in Tibet, considering different heights, latitudes, and longitudes. The analysis includes precipitation
particle size distribution, particle landing speed, precipitation particle number density, and rainfall
intensity at the end. Additionally, the relationship between Z−I distribution and rainfall rate, precipitation
particle distribution fitting, and analysis of Gamma distribution μ−λ parameters for the precipitation
raindrop spectrum characteristics at the four stations are examined. A comparison is made between the
data from the four stations on the Qinghai−Tibet Plateau and some non−plateau areas. Simultaneously,
the analysis of raindrop spectrum data at the Naqu station reveals certain similarities with previous
studies (indicating convective cloud as the primary precipitation at Naqu station). However, some
differences are noted, such as the mean spectral width of convective precipitation at the Naqu station
being relatively narrow.
The relationship between precipitation particle size and particle landing velocity at the four stations



314 indicates that the falling velocity of the four stations essentially coincided when the particle size was less

315 than 1.5 mm. For particle sizes greater than 1.5 mm, the final falling velocity of particles at the four

316 stations is faster at medium and low altitudes than at high altitudes. This is attributed to instruments at

317 high altitudes being closer to the clouds. At the four stations, the proportion of precipitation raindrop

318 spectral particle size less than 1 mm exceeded 91%, and the contribution rate of precipitation was more

319 than 51%. The characteristics of convective cloud precipitation over the Tibetan Plateau exhibit

320 peculiarities that differ from the raindrop spectrum characteristics in the low−altitude areas of the

321 mainland.

322 The six microphysical characteristic parameters at the four stations all increased with altitude, showing

323 a positive correlation with altitude. Regarding the fitted Z−I relationship, the fitting parameter a at the

324 high−altitude station is significantly smaller, while the fitting parameter b is larger. The particle spectrum

325 of high−altitude stations is broader, with a larger equivalent diameter, and the reflectivity of high−altitude

326 stations is significantly higher than that of low−altitude stations.

327 The concentration of small raindrops (less than 1 mm) in the raindrop spectrum of high−altitude stations

328 on the Tibetan Plateau was higher. Both the M−P distribution and the Gamma distribution exhibit good

329 fitting effects for low−altitude stations. Overall, the M−P fit performed better. In the relationship between

330 the $\mu$ and $\lambda$ of the two parameters in the Gamma distribution, the larger the $\mu$, the larger the weighted

331 average diameter of the mass when the $\lambda$ remains constant. In other words, the greater the $\mu$, the greater

332 the precipitation intensity when $\lambda$ remains unchanged.

333 **Data Availability Statement**

334 The data used to support the findings of this study are available from the corresponding author upon

335 request.

336 **Author Contributions**

337 Conceptualization, F.W. and G.C.; methodology, F.W. and Q.W.; software, Y.H. and Q.W.; writing—

338 review and editing, F.W., Y.H. and Y.C.; resources,T.Z. and J.L.; supervision, T.Z. and G.C. All authors

339 have read and agreed to the published version of the manuscript.



**Competing interests**

The contact author has declared that none of the authors has any competing interests.

**Disclaimer**

Publisher's note: Copernicus Publications remains neutral with regard to jurisdictional claims made in the text, published maps, institutional affiliations, or any other geographical representation in this paper. While Copernicus Publications makes every effort to include appropriate place names, the final responsibility lies with the authors.

**Acknowledgements**

We thank the Tibet Meteorological Bureau for the raindrop spectrum data, and the students and teachers of Chengdu University of Information Technology for their help.

**Financial support**

This research was funded by the Open Fund project for Key Laboratory of Land Surface Process and Climate Change in Cold and Arid Regions, Chinese Academy of Sciences (LPCC2020009), and the Natural Science Foundation of Sichuan Province (2022NSFSC0208) and National Natural Science Foundation of China (42075001).

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
