# Peer review of "Distribution characteristics of summer precipitation"

_EGUsphere, 2024_

## Author Response (AR1)

Dear reviewer:

Thank you for your decision and constructive comments on my manuscript.

We agree with the reviewers' suggestions and will incorporate the recommended changes into the manuscript. The comments have been revised in the manuscript. The following is a related question reply:

22 In the Abstract, the "six features" are not defined and only become apparent after reading the main body of the text.
Definitions have been added (average diameter (Dm), mean volume diameter (Dv), mode diameter (Dd), dominant diameter (Dp), and median diameter (Dnd)) 24
27 "better effect" is ambiguous; how about "distribution exhibits a better fit to observations" ?
This refers to the fact that the results obtained using Exponential fitting are closer to the actual observed values. 34
37. if convection is "severe", shouldn't it be a cumulonimbus 100% of the time?
I've deleted it 44
65 "examine" should be "examined"
I have changed it to examined. 74
101. final falling velocity is more commonly known as "terminal" velocity
I have changed it to terminal 108
115 should be "spectra exhibit"
I have changed it to spectra exhibit 125
117, 119 do you perhaps mean "evaluation" instead of "elimination" ?
Evaluate first, and then eliminate those that do not meet the requirements.
121 "deformation" used twice in same sentence, so perhaps a synonym could be substituted for one; Battaglia reference needs the publication year
I've deleted it 131
179-180 says that the mean value of the rain intensity is proportional to the intensity, which doesn't make sense and needs to be rewritten
I've rewritten it The samples from the four stations in the range of $0.5-5$ mm・$h-1$ were the largest, and the obtained standard deviation values were all very small. This indicates a high consistency in rain intensity distribution under weak rain intensity. 190
209 the altitude effect on fallspeed can be calculated, and it would be nice to see the numbers; I think "increased" is meant, not "decreased"
I've rewritten it 223
212 I didn't understand the phrase starting with "leading to detection...."; could you rewrite this for clarification?
I've rewritten it
231 the statement "small particles .... are smaller" : is evidence of this given somewhere in the text? I didn't see it.

232-237 The sentence starting "In contrast" refers to Shigatse, I believe, which is being compared to Zheng'an and Hainan. That could be rewritten for clarification.

242    Dm is mean diameter

I've rewritten it    279

243 "median" is meant instead of "medium", I assume

I've rewritten it    280

244. Suggestion:    if Tables 5 and 6 were ordered from highest to lowest altitude, it would help the reader follow the analysis in the text.

259    intensity I should have 1.4 as its exponent

I've rewritten it    316

276    by "larger change", what is changing?    Or do you mean the maximum (or mean) raindrop size is larger?

Yes, my means the mean raindrop size is larger. This sentence has been rewritten.

277 by "raindrop size", do you mean to say "mean raindrop size"?

I've rewritten it 336

296 Reference is made to Eq. 11, but I don't see that equation in the text.

The Eq.11 is on line 176.

302    should be "light and moderate", not just "moderate"

I've rewritten it 361

317 you state "this is attributed to instruments at    high altitudes being closer to the clouds".    I don't understand what is meant by that.    All drops should accelerate to terminal velocity within several meters of fall.

I've deleted it    377

325    the spectrum is said to be "broader", but where is that shown?

This is reflected in the microscopic parameters of raindrop size.

329 "slightly better", as you said in the Abstract.

Changed to slightly better

1.Lines 103 -114 and Equation (1). The terminal fall speed presented in Atlas et al. (1973) and shown in Equation (1) is for raindrops falling at sea level. The terminal fall speed is dependent on altitude, with raindrops falling faster at higher altitudes. Table 2 in Atlas et al. (1973) list fall speed corrections versus altitude in a Standard Atmosphere. Also, the work of Foote and Du Toit (1969, Journal of Applied Meteorology, pages 249-253) provides a fall speed adjustment for observations made at altitudes above sea level. The discussion of air density adjustment to terminal fall speeds needs to be included in the manuscript.

I have revised it and taken into account the effect of air density. 112

2. Line 206-210 and Figure 3. The raindrop terminal fall speed at elevations over 3000 above sea level are faster than raindrop terminal fall speeds near sea level. Can this difference shown in Figure 3 be described by the expected air density adjustments suggested by Foote and Du Toit (1969)? Is there a better air density adjustment that fits these data?

I have revised it and taken into account the effect of air density. 223

3. Lines 259-262, and Figure 4. Given that the raindrops fall faster at higher altitudes, it is expected that rain with the same radar reflectivity factor will have larger rainfall rates at 3000 m elevation than at sea level. Thus, it is to be expected that the comparison with the sea level Z-I relationships (Z = 300 I^1.4) will underestimate the observed Z-I relationship at sites above 3000 m. The manuscript needs to address the challenge of comparing observations above 3000 m with previous work done at sea level. This could be done by adjusting the sea level relationships to elevation or adjusting the elevation data to sea level. Both ways have their advantages and disadvantages. The manuscript needs to reconcile the altitude differences.

4. Lines 242-243. What are the equations for average volume diameter (Dv), mode diameter (Dd), dominate diameter (Dp), and medium diameter (Dnd)? I would like to estimate these quantities in my disdrometer data, so I would like to see the equations in the body or appendix of the manuscript.

5. Lines 243-246. And Table 5. Since the mass-weighted diameter is related to rain intensity, the simple comparison of mass-weighted diameter (Dm) between sites is not very informative in describing the different microphysical characteristics occurring at the different sites. The analysis should partition the data based on rain intensity. For example, compare the Dm for limited rain rate intensity intervals shown in Table 2. Thus, this analysis would ask the question: for a given rain rate, how does Dm change between the sites?
I have revised it 279
Minor suggestions to improve the manuscript

1. Line 14, and elsewhere, the use of the word "particle" could refer to snow particles or raindrops. If the measurement is ambiguous and it is not known whether the particle is made of frozen or liquid water, then the use of 'particle' is appropriate. On the other hand, if only liquid drops or droplets are studied, then the manuscript will be easier to read if the word "particle" is replaced with "drop" or "droplet".

2. Lines 25-26, and elsewhere. First, the "M-P distribution" is not defined and the reference to Marshall and Palmer (1948) is not referenced. Please include the reference. Second, the "M-P" distribution is different than a general exponential distribution because the 'M-P distribution' defines the coefficients in the exponential distribution (and presented in their 1948 paper). You cannot fit parameters to the M-P distribution (as suggested on line 25-26) When non-M-P parameters are estimated and used in an exponential distribution, then it is not called a M-P distribution, it is called an exponential distribution. Please clarify the text.
I have revised it 170
3. Line 27. What does it mean for a fitting to exhibit "a slightly better effect"? Does this mean a smaller cost function? Please clarify.
This refers to the fact that the results obtained using m-p fitting are closer to the actual observed

values.

4. Line 33. I have not heard before that different precipitation processes can change the "ground heating effect." What is this ground heating effect and how is it measured? Is it measured from space or from air temperature gauges? A reference would also be helpful.

5. Line 102, and elsewhere. The "final falling velocity" is usually called the "terminal velocity". Please change this phrase in the manuscript.

I have revised it 108

6. Lines 121-125. These lines define an axis ratio that is not used in the rest of the manuscript. Please omit this text.

I've deleted it

7.Line 207. Where is Guizhou located? Is it at sea level?

Guizhou is located in the southeast of southwest China, between 103°36'-109°35' E and 24°37'-29°13' N, with an average elevation of 1100m.